# Airway Foreign Body Mimicking an Endobronchial Tumor Presenting with Pneumothorax in an Adult: A Case Report

**DOI:** 10.3390/medicina57010050

**Published:** 2021-01-08

**Authors:** Jun-Ho Ha, Byeong-Ho Jeong

**Affiliations:** 1Department of Medicine, Samsung Medical Center, Sungkyunkwan University School of Medicine, Seoul 06351, Korea; junho91.ha@samsung.com; 2Division of Pulmonary and Critical Care Medicine, Department of Medicine, Samsung Medical Center, Sungkyunkwan University School of Medicine, Seoul 06351, Korea

**Keywords:** pneumothorax, airway foreign body, bronchoscopy

## Abstract

Foreign body (FB) aspiration occurs less frequently in adults than in children. Among the complications related to FB aspiration, pneumothorax is rarely reported in adults. Although the majority of FB aspiration cases can be diagnosed easily and accurately by using radiographs and bronchoscopy, some patients are misdiagnosed with endobronchial tumors. We describe a case of airway FB that mimicked an endobronchial tumor presenting with pneumothorax in an adult. A 77-year-old man was referred to our hospital due to pneumothorax and atelectasis of the right upper lobe caused by an endobronchial nodule. A chest tube was immediately inserted to decompress the pneumothorax. Chest computed tomography with contrast revealed an endobronchial nodule that was seen as contrast-enhanced. Flexible bronchoscopy was performed to biopsy the nodule. The bronchoscopy showed a yellow spherical nodule in the right upper lobar bronchus. Rat tooth forceps were used, because the lesion was too slippery to grasp with ellipsoid cup biopsy forceps. The whole nodule was extracted and was confirmed to be a FB, which was determined to be a green pea vegetable. After the procedure, the chest tube was removed, and the patient was discharged without any complications. This case highlights the importance of suspecting a FB as a cause of pneumothorax and presents the possibility of misdiagnosing an aspirated FB as an endobronchial tumor and selecting the appropriate instrument for removing an endobronchial FB.

## 1. Introduction

Foreign body (FB) aspiration usually occurs in children, and adult cases account for only 20% of the total reported incidents [1,2]. Pneumothorax is an uncommon presentation of an airway FB in children and is very rare in adults [3,4]. Although the majority of cases can be diagnosed correctly in the early period, some patients can be misdiagnosed with upper respiratory tract infections, pneumonia, laryngitis, and with asthma in children [5,6] or chronic obstructive pulmonary disease, asthma, sputum impaction, and suspicious lung cancer in adults [7]. Once an accurate diagnosis of airway FB is determined in adults, the FB can be successfully extracted using flexible bronchoscopy, and this approach is associated with a success rate of 61–100% [8]. However, it is important to use the appropriate equipment to increase the success rate [9,10]. Herein, we describe a case of an airway FB that mimicked an endobronchial tumor presenting with pneumothorax in an adult, which was successfully removed under flexible bronchoscopy with rat tooth grasping forceps.

## 2. Case Presentation

A 77-year-old man was referred to our hospital due to pneumothorax and atelectasis of the right upper lobe caused by an endobronchial nodule. Due to his advanced age, low performance status, and comorbidities (such as chronic kidney disease that did not require dialysis and coronary artery disease), he resided in a nursing hospital. Five days before admission to our hospital, the patient reported sudden chest pain and dyspnea. The initial chest radiography and computed tomography (CT) revealed a right pneumothorax, pleural effusion in the same side, and a round 9-mm-sized nodule in the right upper lobar bronchus combined with atelectasis (Figure 1). A chest tube was immediately inserted to decompress the pneumothorax. A chest CT with contrast was taken for characterization of the endobronchial nodule while hemodialysis was initiated. The endobronchial nodule was seen as contrast-enhanced (Figure 1D). The endobronchial nodule was observed on a flexible bronchoscopy, but biopsy of the lesion failed. On the fifth day of hospitalization, the patient was referred to our hospital for further evaluation and management.

On arrival, his vital signs were as follows: blood pressure 143/67 mmHg, pulse rate 71/min, respiratory rate 18/min, and body temperature 36.3 °C. The chest tube was well-functioning, without continuous air leakage. Although the pleural lactate dehydrogenase level (351 U/L) was high, its appearance was serous, and the serum-pleural fluid protein gradient was 3.3 g/dL (5.2 minus 1.9 g/dL). Therefore, pleural effusion can be considered a transudate.

Flexible bronchoscopy (EVIS BF 1T260; Olympus Co., Tokyo, Japan) was performed to biopsy the nodule. The bronchoscopy showed a yellow spherical nodule in the right upper lobar bronchus (Figure 2). The lesion was too slippery to biopsy with ellipsoid cup biopsy forceps (FB-21C-1, Olympus Co.). Additionally, the lesion seemed to slightly rotate in place. Since the issue seemed to be an airway FB rather than a lesion fixed in the bronchus, we used rat tooth grasping forceps for extraction (FG-26C-1, Olympus Co.). When grasping the nodule with rat tooth forceps, the whole nodule was extracted without any persisting endobronchial mucosal lesion. The nodule was confirmed to be a FB and was determined to be a green pea. However, the patient denied the experience of food aspiration. The chest tube was removed after the procedure, and a chest radiography showed that the atelectasis and pneumothorax were resolved (Figure 3). One month after discharge, the patient visited the outpatient clinic and was in good condition, without any symptoms.

## 3. Discussion

In this case study, the patient presented with chest pain, and his chest CT revealed a right pneumothorax, atelectasis of the right upper lobe and an endobronchial nodule in the right upper lobar bronchus. Since the nodule was visualized as contrast-enhanced in CT images, it was initially considered to be an endobronchial tumor, such as a carcinoid. The first attempt to biopsy using standard forceps was not successful. In the next attempt, the whole nodule was extracted using rat tooth forceps and was confirmed to be a green pea that was mimicking an endobronchial tumor. However, this patient said that he never suffered aspiration while eating. We think that the aspiration symptom was minimal or absent due to his elderly and poor general condition. After removal of the FB, atelectasis and pneumothorax were completely resolved.

The overall incidence of FB aspiration has a bimodal distribution, with one peak at one to two years of age and the other peak at over 60 years of age [1,11]. However, FB aspiration usually occurs in children, and adult cases account for only 20% of the total [1,2]. Airway FBs are rarely located in the upper lobar bronchi under the influence of gravity. In this case, the patient was bedridden all day, so the airway FB was able to be located in the upper lobar bronchus. Clinical manifestations of FB aspiration can vary, depending on the size of the FB and the location where it becomes lodged [1,3,10]. In addition, atelectasis, pneumonia, respiratory distress, bronchiectasis, cardiopulmonary arrest, and pneumothorax can occur as complications of airway FB [3,4]. Of these, the incidence of pneumothorax is reported to be 0.5–1.9% in children but is extremely rare in adults [3]. Two mechanisms have been proposed to explain the development of pneumothorax. The first mechanism is that FB in the airway behaves as a check valve [4,12,13]. The FB causes air trapping, which can lead to increased alveolar pressure and rupture of the alveolar membrane. The second mechanism is related to mechanical disruption of the mucosal membrane of the airway [4]. The loss of mucosal membrane integrity leads to pneumomediastinum and pneumothorax. The different incidences of pneumothorax in adults and children may be associated with the airway FB location. More proximal tracheal airway obstruction occurs more frequently in the younger pediatric population because of their smaller airway diameters and usually leads to more severe symptoms [1,8,9]. Reports have noted that exertion could promote the onset of pneumothorax [14]. The more severe the symptom, the more forcefully inhalation and exhalation can result in markedly elevated pressure. Less severe symptoms in adults might explain why pneumothorax-related aspirated FB occurs less frequently in adults than in children.

Among many imaging modalities, CT has become the gold standard for imaging studies when FB aspiration is suspected [10]. The radiologic manifestations of FB aspiration include either direct visualization of the FB or indirect signs in the form of pneumonia, atelectasis, hyperinflation, or localized bronchiectasis [1,9,10,15]. Although chest CT can diagnose airway FB with high sensitivity and specificity, some cases of FB that mimic endobronchial tumors have been reported [15,16,17]. The diagnosis of FB aspiration by imaging can be challenging, because the majority of foreign bodies are radiolucent [1,10,15], and other indirect radiologic signs of FB can be seen in other endobronchial tumors [18]. Even in positron emission tomography images, inflammation surrounding FB, which can result in elevated ^18^F-fluorodeoxyglucose uptake, make it difficult to confirm a differential diagnosis with tumors [16,17]. Bronchoscopy is useful not only for extraction of a FB but, also, for diagnosing FB by directly visualizing the lesion. However, inflammatory changes with granulomas or necrotic debrides induced by aspirated FB can be similar to a tumor, even in bronchoscopy inspections [15,19]. In this case, the nodule appeared as a well-defined spherical mass and also appeared to have associated contrast enhancement. These findings in CT imaging suggest tumors, like carcinoids; however, the nodule was identified as a green pea. Ultimately, a comprehensive approach including suspicion, a detailed history of aspiration, images, and bronchoscopy is needed to accurately diagnose aspirated FBs.

An airway FB in an adult can be successfully removed with either flexible or rigid bronchoscopy [8,9,10,15]. A variety of instruments are used during flexible bronchoscopy to extract an airway FB [9,10]. Forceps are the most frequently utilized tool for FB removal [10]. Forceps are available in multiple configurations of teeth and bite diameters to accommodate objects of various sizes and textures. Shark tooth, rat tooth, alligator tips, and rubberized tips can be used, depending on the size and texture of the airway FB. Standard cup forceps used for endobronchial biopsy are not as useful as the previously listed forceps for removing an airway FB [10]. Even in this case, the FB was too slippery and large to grasp with cup forceps. Consequently, the tooth size and larger opening width of the rat tooth forceps allowed for better grasping of the airway FB. It is important to select the appropriate instruments, depending on the characteristics and location of the FB.

## 4. Conclusions

In summary, our report highlighted some learning points: (1) it is important to suspect a FB as a cause of secondary spontaneous pneumothorax, (2) aspirated FB can mimic an endobronchial tumor, and (3) selection of the appropriate instrument is important for successful extraction of an airway FB.

## Figures and Tables

**Figure 1 medicina-57-00050-f001:**
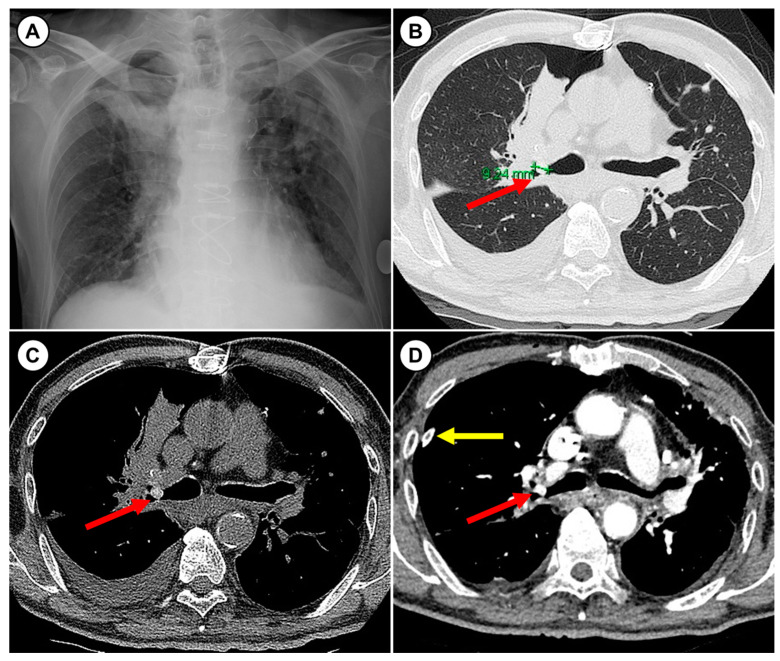
Chest radiography and computed tomography before bronchoscopy. (**A**) Chest radiography obtained at the time of the first visit due to chest pain showed pneumothorax with atelectasis of the right upper lobe. (**B**) Chest computed tomography without a contrast agent showed a 9-mm-sized round endobronchial nodule (red arrow) in the right upper lobar bronchus, a small amount of pneumothorax with atelectasis of the right upper lobe, and a small amount of pleural effusion in the right pleural cavity. (**C**) On the mediastinal window, the endobronchial nodule (red arrow) area had high Hounsfield units (mean = 112). (**D**) Chest computed tomography images with contrast agent were obtained after chest tube insertion (yellow arrow). The endobronchial nodule appeared as a contrast enhancement (mean Hounsfield units = 225).

**Figure 2 medicina-57-00050-f002:**
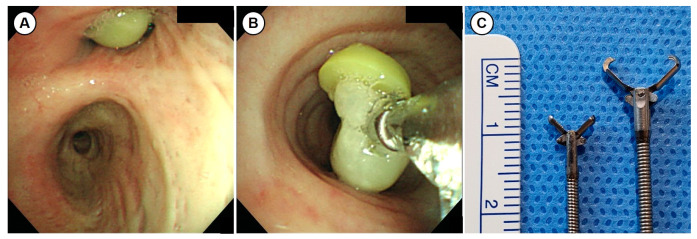
Bronchoscopy images. (**A**) A yellow endobronchial nodule was seen in the right upper lobar bronchus and was too slippery to grasp with biopsy forceps. (**B**) The endobronchial nodule was en bloc extracted with rat tooth forceps. The nodule was determined to be a green pea vegetable. (**C**) A photo of commonly used biopsy forceps (ellipsoid cup biopsy forceps, FB-21C-1, Olympus Co., left side) that failed a biopsy and of forceps (rat tooth grasping forceps, FG-26C-1, Olympus Co., right side) that successfully removed an airway foreign body.

**Figure 3 medicina-57-00050-f003:**
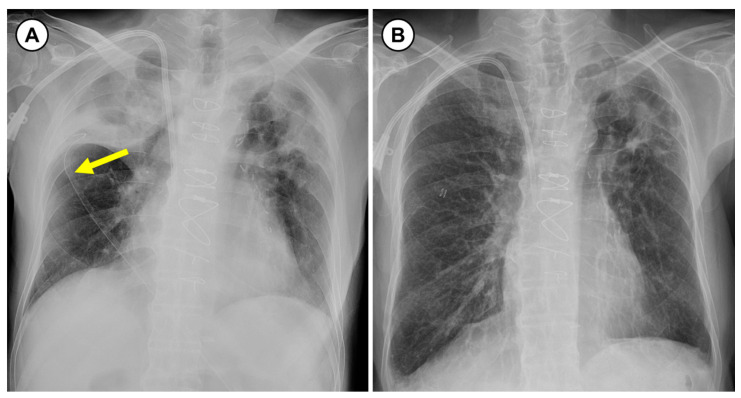
Chest radiography before and after bronchoscopy. (**A**) On chest radiography before bronchoscopy, a chest tube (yellow arrow) was inserted in the right thoracic cage. The pneumothorax improved, but there was still volume loss owing to right upper lobe atelectasis. (**B**) One day after bronchoscopy, the chest tube was removed, and the atelectasis completely disappeared.

## Data Availability

Data and material are available on reasonable request.

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
