# Peer review of "Airway Foreign Body Mimicking an Endobronchial Tumor Presenting with Pneumothorax in an Adult: A Case Report"

_medicina, 2021, doi:10.3390/medicina57010050_

Round 1
Reviewer 1 Report
This is a case report that the patient with foreign body aspiration manifested pneumothorax and atelectasis. It is relatively rare to cause aspiration in the right upper bronchus rather than the lower lobe bronchus. The aspirated material occluded the airway and caused atelectasis, which was mimicking pneumothorax.
Comments
- What was the initial differential diagnosis of this patient? Did the patient eat something just before complaint symptoms?
- What was the route of chest drainage to the air space? Did they insert from the anterior chest wall?
- Chest CT showed complete atelectasis of the right upper lobe. Did they observe any air leakage during chest drainage? What were the findings of pleural effusion?
- In Figure 3, did the authors insert a CV line from the right subclavicular vein? If so, what was an indication?
- Was it pneumothorax? If the right upper lobe volume was large enough, it might cause airspace by complete atelectasis.
- Did the authors firstly suspect an endobronchial tumor for acute onset disease?
- Please add the image of rat-tooth forceps instead of Figure 2C.
Author Response
This is a case report that the patient with foreign body aspiration manifested pneumothorax and atelectasis. It is relatively rare to cause aspiration in the right upper bronchus rather than the lower lobe bronchus. The aspirated material occluded the airway and caused atelectasis, which was mimicking pneumothorax.
Comments
C1. What was the initial differential diagnosis of this patient? Did the patient eat something just before complaint symptoms?
R1. Thank you for your thoughtful comments to much improve our report. In the previous hospital and our hospital, the initial differential diagnosis of this patient was an endobronchial tumor on the RUL bronchus with atelectasis of RUL and pneumothorax. As written in Line 51-52 and 62-64, the endobronchial nodule was seen as contrast enhanced, so we had no choice but to suspect the tumor such as carcinoid tumor (as written in Line 97-98).
Eventually, after confirming that the endobronchial nodule was a foreign body, not a tumor, the patient was asked if he had any experience of food aspiration. But, this patient denied the experience of food aspiration. We think that the aspiration symptom was minimal or absent due to the elderly and poor general condition.
We added the patient information in Line 77-78 - “However, the patient denied the experience of food aspiration” and in Line 100-102 – “However, this patient said that he had never suffered aspiration while eating. We think that the aspiration symptom was minimal or absent due to the elderly and poor general condition.”
C2. What was the route of chest drainage to the air space? Did they insert from the anterior chest wall?
R2. Yes. In the previous hospital, the chest tube was inserted through the anterolateral chest wall (between 2nd and 3rd rib).
C3. Chest CT showed complete atelectasis of the right upper lobe. Did they observe any air leakage during chest drainage? What were the findings of pleural effusion?
R3. This patient transferred to our hospital 6 days after chest tube insertion. In our hospital, there was no active continuous air leakage through the chest tube, but some bubbles and oscillation was observed (as written in Line 66-67). There are no records of previous hospitals, but these records of our hospital are kept in the nursing records.
In our hospital, the laboratory findings of pleural effusion was as follows: protein 1.9 g/dL, LDH 351 U/L, glucose 111 mg/dL. Serum protein level was 5.2 g/dL. Although pleural LDH was high, protein gradient was greater than 3.1 g/dL. So, pleural effusion can be considered a transudate. We added the patient information as follows (Line 67-69): “Although pleural lactate dehydrogenase level (351 U/L) was high, its appearance was serous and the serum–pleural fluid protein gradient was 3.3 g/dL (5.2 minus 1.9 g/dL). Therefore, pleural effusion can be considered a transudate.”
C4. In Figure 3, did the authors insert a CV line from the right subclavicular vein? If so, what was an indication?
R4. As written in the manuscript, the patient had underlying diseases (chronic kidney disease that did not require dialysis and coronary artery disease). However, at this event, this patient needed dialysis (as written in Line 51), so permanent dialysis catheter (permcath) was inserted in the previous hospital.
C5. Was it pneumothorax? If the right upper lobe volume was large enough, it might cause airspace by complete atelectasis.
R5. On simple chest radiography and chest CT, there is air in the pleural space. Although we have no CT image before this event or after the cure of this event, old tuberculosis scars (without treatment history) are seen on the opposite lung (left apex) and small bullae are seen on both lung apexes. So, we think this case is a pneumothorax case “caused by rupture of bullae and/or old tuberculosis scars”, caused by an airway FB and atelectasis.
(Coronal view CT image before chest tube insertion. Multiple bullae are shown on both lung apexes (red arrows). Old tuberculosis scars are shown on the left apex (blue arrow).)
(Coronal view CT image before chest tube insertion. Air spaces are shown on apex (red arrow), between upper lobe and middle lobes (yellow arrow), and between middle and lower lobes (blue arrow).)
C6. Did the authors firstly suspect an endobronchial tumor for acute onset disease?
R6. Our hospital is performing the most bronchoscopic intervention for endobronchial tumors in South Korea.
([1] Shin et al. BMC Pulmonary Medicine (2018) 18:46 - Interventional bronchoscopy in malignant central airway obstruction by extra-pulmonary malignancy; [2] Kim et al. BMC Pulmonary Medicine (2020) 20:54 - Prognostic factors for survival after bronchoscopic intervention in patients with airway obstruction due to primary pulmonary malignancy; [3] Kim et al. Lung Cancer (2020) 146:58-65 - Clinical outcomes and the role of bronchoscopic intervention in patients with primary pulmonary salivary gland-type tumors)
Although the endobronchial tumors themselves usually do not grow rapidly, symptoms can develop suddenly if the endobronchial tumors are located on the peripheral airway (because obstruction of peripheral airway alone may not cause symptoms such as dyspnea and cough, but after further progression, pneumonia and atelectasis may occur suddenly and manifest as acute symptoms).
C7. Please add the image of rat-tooth forceps instead of Figure 2C.
R7. We changed the Figure 2C and its Figure legend (line 85-88) as follows: (C) A photo of commonly used biopsy forceps (ellipsoid cup biopsy forceps, FB-21C-1, Olympus Co., left side) that failed a biopsy and of forceps (rat-tooth grasping forceps, FG-26C-1, Olympus Co., right side) that successfully removed an airway foreign body.

Reviewer 2 Report
The case report authored by Jun-Ho Ha and Byeong-Ho Jeong (manuscript ID “medicina-1050446”) describes a case of airway foreign body (a green pea) that mimicked an endobronchial tumor presenting with pneumothorax in a 77-year-old man. The authors reported that after removal of the whole nodule under flexible bronchoscopy with rat-tooth grasping forceps, atelectasis and pneumothorax were completely resolved. This work is interesting and well-written. I have some minor comments that could help to improve the relevance of this manuscript.
1. The authors should discuss the novelty of the management of their clinical case in more detail.
2. P1, L42: Please, write “Case presentation” in bold and italic free.
3. P4, L144: Please, insert “and” before “(3) selection”.
Author Response
The case report authored by Jun-Ho Ha and Byeong-Ho Jeong (manuscript ID “medicina-1050446”) describes a case of airway foreign body (a green pea) that mimicked an endobronchial tumor presenting with pneumothorax in a 77-year-old man. The authors reported that after removal of the whole nodule under flexible bronchoscopy with rat-tooth grasping forceps, atelectasis and pneumothorax were completely resolved. This work is interesting and well-written. I have some minor comments that could help to improve the relevance of this manuscript.
C1. The authors should discuss the novelty of the management of their clinical case in more detail.
R1. Thank you for the thoughtful comments. We added in the discussion section as follows: “However, this patient said that he had never suffered aspiration while eating. We think that the aspiration symptom was minimal or absent due to the elderly and poor general condition. (Line 100-102)” and “Airway FBs are rarely located in the upper lobar bronchi under the influence of gravity. In this case, the patient was bed ridden all day, so the airway FB was able to be located in the upper lobar bronchus. (Line 106-108)”, we changed the Figure 2C and its Figure legend to enhance the importance of choosing forceps as follows: “A photo of commonly used biopsy forceps (ellipsoid cup biopsy forceps, FB-21C-1, Olympus Co., left side) that failed a biopsy and of forceps (rat-tooth grasping forceps, FG-26C-1, Olympus Co., right side) that successfully removed an airway foreign body. (Line 85-88)”.
C2. P1, L42: Please, write “Case presentation” in bold and italic free.
R2. We changed the font as you said (Line 42).
C3. P4, L144: Please, insert “and” before “(3) selection”.
R3. We inserted the word as you said (Line 154).

Round 2
Reviewer 1 Report
The queries were addressed.